# Nucleic Acid Aptamer-Based Biosensors: A Review

**DOI:** 10.3390/biomedicines11123201

**Published:** 2023-12-01

**Authors:** Beatriz Sequeira-Antunes, Hugo Alexandre Ferreira

**Affiliations:** 1Institute of Biophysics and Biomedical Engineering, Faculty of Sciences, University of Lisbon, Campo Grande, 1749-016 Lisboa, Portugal; 2Exotictarget, 4900-378 Viana do Castelo, Portugal; 3Instituto de Engenharia de Sistemas e Computadores-Microsistemas e Nanotecnologias (INESC-MN), 1000-029 Lisbon, Portugal

**Keywords:** biomarkers, aptamers, SELEX process, aptasensors, electrical aptasensors, optical aptasensors, magnetic aptasensors, point of care

## Abstract

Aptamers, short strands of either DNA, RNA, or peptides, known for their exceptional specificity and high binding affinity to target molecules, are providing significant advancements in the field of health. When seamlessly integrated into biosensor platforms, aptamers give rise to aptasensors, unlocking a new dimension in point-of-care diagnostics with rapid response times and remarkable versatility. As such, this review aims to present an overview of the distinct advantages conferred by aptamers over traditional antibodies as the molecular recognition element in biosensors. Additionally, it delves into the realm of specific aptamers made for the detection of biomarkers associated with infectious diseases, cancer, cardiovascular diseases, and metabolomic and neurological disorders. The review further elucidates the varying binding assays and transducer techniques that support the development of aptasensors. Ultimately, this review discusses the current state of point-of-care diagnostics facilitated by aptasensors and underscores the immense potential of these technologies in advancing the landscape of healthcare delivery.

## 1. Introduction

Biosensor technology has emerged as a dynamic and rapidly evolving field, responding to the pressing need for precise, rapid, and cost-effective measurements to address challenges related, for example, to healthcare, environmental monitoring, and food safety [1]. These challenges encompass issues like the need for laboratory facilities and prolonged response times.

A biosensor can be described as a system that combines a biological recognition element with transducers [2]. This integration enables the identification, measurement, and communication of data concerning the presence or concentration of distinct biological molecules or chemical compounds in a given sample [3,4]. Essentially, a biosensor converts a biochemical interaction into a measurable physical signal [1]. The mechanisms underlying these transduction processes can use a diverse array of technologies, including optical [5], electrochemical [3], piezoelectric [6], and magnetic [7].

The linchpin of biosensors lies in their recognition molecules, whose sensitivity and specificity wield substantial influence over the efficacy of these sensing devices [1]. Antibodies are frequently used as the biological recognition element; however, production of antibodies requires experimentation with animals, which is a costly and time-consuming process, besides raising ethical concerns [8]. In contrast, aptamers, both nucleic-acid-based aptamers and peptide aptamers [9], have emerged as a compelling alternative, featuring versatility in their molecular recognition capabilities, since aptamers can be selected against a wide range of targets, as well as offering precision and ease of chemical synthesis and modification, having therefore lower production costs and being more ethically acceptable [10,11]. Owing to these unique properties, aptamer-based biosensors, often referred to as aptasensors, have assumed critical roles in clinical diagnostics [1,12], disease monitoring [13], biomarker identification [14], and the analysis of diverse sample types, e.g., peripheral blood and serum [4,15,16]. Aptasensors, with their versatility and high specificity, find applications beyond the biomedical realm, such as in environmental monitoring [17], food safety [18], and agriculture [19].

In this paper, we aim to delve into the advantages of aptamers over traditional antibodies as biological recognition elements in biosensors, alongside an exploration of conventional signal transduction strategies. Furthermore, this paper aims to spotlight the latest advancements in aptamer-based biosensors within healthcare and clinical diagnostics while addressing the associated opportunities and challenges that lie ahead. Overall, this review paper aligns with the increasing attention directed towards aptamers, positioning them as integral elements in biosensing.

## 2. Nucleic-Acid-Based Aptamers: Nature’s Molecular Recognition Tools

Aptamers typically are short nucleic acid molecules, composed of DNA, RNA, or alternatively peptide molecules, that are specifically designed in the laboratory to bind with high affinity and specificity to a target molecule, such as a small molecule, protein, nucleic acids, or even a whole cell [4,10,20,21].

### 2.1. Nucleic-Acid-Based Aptamers

The journey of aptamers began in the early 1990s when researchers presented a method of in vitro generation of high-affinity molecules against selected targets. This pioneering work led to the successful selection of the first aptamer in 1990, which was designed to target T4 DNA polymerase [22]. The term “aptamer” itself, derived from the Latin *aptus* meaning “to fit” and *meros* meaning “part”, was coined by researchers Andrew Ellington and Jack Szostak [23]. Aptamers gained significant recognition when the U.S. Food and Drug Administration (FDA) granted approval for Pegaptanib in 2004, marking the first-ever aptamer-based therapeutics to treat age-related macular degeneration [24].

Nucleic-acid-based aptamers are developed through a process called Systematic Evolution of Ligands by Exponential enrichment (SELEX), as outlined in Figure 1 [10,25]. SELEX comprises a series of iterative steps, involving selection, elution, and amplification, with the goal of identifying aptamers that strongly bind to a specified target molecule [26]. The process starts with the creation of a diverse library of nucleic acid molecules, often up to approximately 10^14^ different molecules [26], typically single-stranded DNA or RNA, characterized by random sequences to encompass a wide array of possibilities. Subsequently, this initial library is mixed with the target molecule, and the unbound nucleic acid sequences are separated from those that interacted with the target. The latter group is then eluted and amplified through polymerase chain reaction (PCR) for DNA libraries or reverse-transcription PCR for RNA libraries, leading to the generation of numerous copies of sequences that have demonstrated an affinity for the target. These amplified sequences serve as the foundation for the subsequent selection round, a process that is repeated through multiple rounds, usually ranging from six to fifteen. After multiple selection rounds, the resulting pool of sequences is analysed and subjected to sequencing, and those demonstrating the highest specificity and affinity for the target are singled out and deemed potential aptamers.

The identification of candidate aptamers through the SELEX process represents a pivotal stage in the development of aptamer-based technologies. Additionally, investigation of their binding characteristics, encompassing assessments under differing conditions such as temperature, pH, and ionic strength, is essential for understanding their stability and adaptability across diverse environments.

### 2.2. Aptamers vs. Antibodies

In addition to aptamers, antibodies stand out as one of the frequently employed molecular recognition instruments [8].

Antibodies, also known as immunoglobulins, are Y-shape proteins produced by the immune system in response to the presence of various antigens, encompassing substances like chemicals, medications, bacteria, and other foreign molecules [27].

Some drawbacks have been identified in the production of antibodies, which typically require animals to be used. In particular, antibodies can only be produced against substances that provoke an immune response, resulting in a limited number of targets [28]. Moreover, it is hard to produce large and homogeneous batches for antibodies, due to the variability of immune responses against the same antigens [28].

Antibody denaturation is an irreversible process in which the three-dimensional structure of an antibody is disrupted or altered. This can be caused by various factors such as changes in pH, temperature, and salt concentration. When an antibody undergoes denaturation, it may lose its binding specificity and affinity for its target antigen, which is a limitation [28]. These environment variations may lead to a limited storage time and a reduced antibody shelf life, i.e., their useful lifetime [29].

Aptamers in comparation with antibodies offer some benefits. Since aptamers are developed in vitro and no animals are needed, substances that do not provoke an immune response can be used to produced aptamers. Additionally, because the SELEX process is performed under controlled laboratory conditions, aptamers show reduced batch-to-batch variation. As such, aptamers can be rapidly selected and modified to bind with precision to a wide range of targets, including ions, small molecules, proteins, and cells [10]. Like antibodies, denaturation of aptamers also occurs. However, in the case of aptamers, this process is reversible, making their storage time and shelf life longer [28].

Apart from all these advantages, aptamers also have some drawbacks. In particular, RNA aptamers are prone to degradation by nucleases in biological fluids, which reduces their half-life and effectiveness [21,30]. Additionally, aptamers are selected from a pool of random oligonucleotides that may not cover the full range of possible sequences and structures that can bind to a given target. This may limit the discovery of optimal aptamers with high affinity and specificity [30].

Table 1 summarizes some of the characteristics that make aptamers a more suitable choice than antibodies in various applications such as biosensing and therapeutics.

### 2.3. Aptamers for Diagnostics

In diagnostics, aptamers play a vital role in detecting and quantifying specific biomarkers indicative of diseases, including infectious diseases [33], cancer [34], cardiovascular diseases (CVDs) [35], metabolomic diseases [36], and neurological disorders [37].

Aptamers have been used to target pathogenic microorganisms, such as bacteria and viruses, for the diagnosis of infectious diseases. An example of that is the aptamer developed by Amraee et al. [38] and Yu et al. [39] against *Escherichia Coli* (*E. coli*) O157:H7, which can lead to severe gastroenteritis. Other studies reported aptamers for the detection of pathogenic bacteria, such as *Salmonella typhimurium* [40,41], *Salmonella enteritidis* [41], and *Mycobacterium tuberculosis* [42]. Regarding pathogenic viruses, aptamers were studied to target the avian influenza virus (AIV), the human immunodeficiency virus (HIV) [43,44,45], and, more recently, the severe acute respiratory syndrome coronavirus 2 (SARS-CoV-2) [46].

Regarding cancer biomarkers and antigens, some aptamers have already shown their potential as diagnosis tools. Of these, aptamers sequences for nucleolin [47], tenascin [48], prostate-specific antigen (PSA) [49], mucin 1 (MUC1) [50], annexin A2 [51], matrix metalloprotease-9 (MMP-9) [52,53], and human epidermal growth factor receptor 2 (HER2) [54] were identified.

A variety of cardiac biomarkers have been established which are useful for diagnosis, monitoring of disease, outcome prognosis, and for risk assessment of CDVs. For some of these biomarkers, aptamer sequences have been identified, such as for cardiac troponin I (cTnI) [55,56,57,58,59,60] and troponin T [61,62,63,64], myoglobin [65,66,67,68], and creatinine kinase [69]. Apart from cardiac biomarkers, aptamers based on inflammatory biomarkers have also been developed, due their relevant association with CDVs. Examples of those are the aptamers identified against interleukin-6 [70,71,72,73], C-reactive protein (CRP) [74,75,76,77,78,79], and tumour necrosis factor alpha [80,81,82].

Moreover, aptamers exhibit significant potential in the field of metabolic diseases, such as diabetes mellitus and obesity. Examples of such aptamers targeting metabolic biomarkers include haemoglobin A1C (HbA1c) [83,84], glycated human serum albumin (GHSA) [85], retinol binding protein 4 (RBP4) [86], mature adipocyte cell lines [87], and visceral-adipose-tissue-derived serpin (Vaspin) [88].

In the field of neurology, on the other hand, the use of aptamers has been relatively restricted up to this point, despite their vast potential for diverse applications [37]. Examples of aptamers targeting neurological biomarkers such as Amyloid-β (Aβ) [89], or more specifically Aβ fragment 1-40 [90], Tau proteins 381 [91,92] and 441 [92], dopamine [93], α-synuclein (α-syn) [94], and huntingtin protein [95], have already been developed.

The identification of aptamers tailored to specific biomarkers holds significant promise for advancing diagnostics across a wide spectrum of medical conditions. Table 2 provides a summary of some aptamers identified, as well as their type and dissociation constant (K_d_). This constant quantifies the affinity or the strength of the binding interaction between the aptamer and its target molecule, as the smaller the K_d_, the higher the affinity [96].

## 3. Aptasensors: The Emergence of Aptamers in Sensing Technologies

Aptasensors are innovative biosensing devices that use aptamers as the molecular recognition element. The operation of aptasensors is based on the selective binding of aptamers to their target molecules. When the target molecule is present in a sample, it binds to the aptamer, an event which is transduced into a detectable signal through various physical mechanisms, which can then be quantified and used to determine the presence or concentration of the target molecule in the sample. According to that, aptamer-based biosensors can be divided into several groups, among which the optical and the electrical aptasensors stand out. Figure 2 schematizes the operating principle of an aptasensor.

### 3.1. Binding Configurations on Solid Supports

As aptamers have been chosen to interact with a broad spectrum of targets, encompassing small molecules and large macromolecules like proteins, diverse assay configurations have been designed and reported [100]. These configurations are typically categorized into two groups: single-site binding and dual-site binding, as illustrated in Figure 3.

In the case of small-molecule targets, nuclear magnetic resonance studies have observed that the target tends to be confined within the binding pocket of aptamer structures, leaving limited space for interaction with a second molecule. A single-site binding configuration can also be employed for small molecules [27,100]. This binding arrangement uses aptamers designed to recognize and bind to a particular region on the target molecule with high affinity and specificity.

In turn, when dealing with macromolecular targets, both single-site and dual-site binding approaches can be employed [100]. The dual-site binding method, commonly referred to as the sandwich assay and widely used, relies on having a pair of aptamers that bind to distinct regions of the target molecule [101]. In this assay, the target analyte is sandwiched between these two aptamers. One of them, known as the capture aptamer, allows for the capture of the target analyte, while the other, referred to as the reporter aptamer, which is often conjugated with signalling molecules like fluorophores, enzymes, or nanoparticles, is responsible for generating a detectable signal [27,100]. Capture and reporter aptamers typically have different nucleic acid sequences. However, in certain situations, some targets, may have two identical binding sites, allowing for the use of a single aptamer in the sandwich assay. Furthermore, when there are no two aptamers available that share identical or overlapping binding sites on the target, it becomes feasible to employ an antibody as the reporter [101]. This way, aptamer–analyte–antibody sandwich assay is a hybrid detection method that combines the unique properties of aptamers and antibodies to enhance the sensitivity and specificity of target molecule detection.

Additionally, aptamer switches, a special kind of nucleic-acid-based aptamer, can be used, due their unique ability to change their conformation when binding to their targets [102], as illustrated in Figure 4. This allows for the detection of a wide range of analytes, from ions to small molecules and to large biomolecules, making them useful tools in various applications, including biosensing and molecular diagnostics [103].

### 3.2. Transduction Mechanisms

After the aptamers bind to their targets, the transducer translates the binding event into a detectable and measurable signal. Table 3 list some transducer mechanism used in aptasensors and their advantages and disadvantages.

#### 3.2.1. Optical Detection

Optical aptasensors are a category of biosensors that utilize light-based techniques to detect and quantify the binding events between aptamers and their target molecules. These interactions are manifested through various optical phenomena, including fluorescence, chemiluminescence, surface plasmon resonance (SPR), surface-enhanced Raman scattering (SERS), and colorimetry [25].

Among various optical detection mechanisms, fluorescence-based detection stands out as the most commonly employed sensing technique [104]. This prevalence arises from the ability for aptamers to be easily customized with fluorescent tags [105]. Within this transducer mechanism, a fluorescent tag is affixed to either the aptamer or the target molecule. The binding event between the aptamer and the target molecule induces alterations in fluorescence intensity, either enhancing or quenching it [25]. These alterations can be detected and quantified through fluorescence spectroscopy. Despite the numerous advantages offered by fluorescence aptasensors, including high sensitivity, low detection limits, real-time monitoring, cost-effectiveness, and a broad selection of fluorescence labels for biomolecule tagging [25,104], they may encounter challenges related to background fluorescence, photobleaching of fluorophores, and potential interference from complex sample matrices. In essence, fluorescence-based aptasensors harness the sensitivity and selectivity of aptamers and the versatility of fluorescence to provide powerful tools for detecting and quantifying a wide range of target molecules in various applications.

Apart from fluorescence, alternative and distinct optical detection methods can be used, such as chemiluminescence. In this mechanism, light is emitted during chemical reactions involving a chemiluminescent substrate and an enzyme or catalyst, which serves to amplify biomolecular binding. Usually, these two components are conjugated to either the aptamer or the target molecule [25]. This mechanism presents advantages such as low background signal, enhanced sensitivity for target detection, as well as simplicity, the potential for multiplexing, and real-time monitoring. Nonetheless, some challenges may still arise, including the requirement for chemical substrates and limitations in signal duration [25,105]. Besides that, both SPR and SERS have been included as potential optical detection methods. The first one is a label-free technique that measures the changes in the refractive index of a metal surface layer in response to the interaction between the immobilized aptamer on the surface and the target molecule [25]. The SPR-based detection technique has increased in importance in recent years due to its appealing attributes. These include the ability to perform label-free and real-time detection, with high sensitivity and an easy experimental setup [104,106]; however, this transducer method may be expensive [107]. On the other hand, SERS is based on Raman scattering, which translates light absorption and subsequent energy superimposition of molecular vibrations, resulting in emitted photons of higher or lower frequencies than the absorbed photons [108]. As such, SERS aptasensors make use of signals produced by molecules of enhanced Raman scattering that report the binding to aptamers immobilized on a surface. Typically, this transduction mechanism is combined with other optical techniques, such as colorimetric and fluorescence, to enhance detection performance [25]. Finally, SERS aptasensors’ potential for multiplexing and wide-ranging applications makes them valuable tools in various scientific and analytical contexts.

Furthermore, the colorimetric technique is also used in optical aptasensors. This technique operates based on the principle that binding events between aptamers and target molecules induce changes in the colour or absorbance properties of a solution [27]. Colorimetric transducers can be implemented using different materials and methods, including both gold nanoparticles (AuNPs) and DNAzymes with horseradish peroxidase (HRP)-mimicking activity. When a target analyte binds to the functionalized AuNPs, it can induce aggregation or dispersion of the nanoparticles, leading to a change in their colour. On the other hand, an alternative strategy involves the use of DNAzyme complexes with HRP. In this case, the DNAzyme specifically binds to its target analyte, triggering a conformational change or cleavage, allowing for the release of a chemiluminescent substrate. Subsequently, the HRP enzyme catalyses the oxidation of the chemiluminescent substrate, leading to the emission of light and therefore enhancing a measurable signal [27].

As such, this transducer mechanism involves monitoring the colour changes of a solution by the naked eye or by using a spectrophotometer [25,27]. While colorimetric aptasensors offer simplicity, cost-effectiveness, and ease of interpretation, they may have limitations in terms of sensitivity compared to other optical detection methods, in particular to SERS, potentially the most sensitive of the optical techniques.

Globally, optical aptasensors harness the unique binding properties of aptamers and the power of light-based detection techniques to provide highly sensitive and versatile platforms for detecting a diverse array of target molecules in various applications, including proteins, nucleic acids, small molecules, and pathogens.

#### 3.2.2. Electrical Detection

Electrical aptasensors are biosensors that employ electrical signals to detect and quantify binding events between aptamers and their target molecules [25]. These sensors are designed to transduce the binding interactions into measurable electrical signals, allowing for the sensitive detection of various analytes [107]. Electrical aptasensors can utilize different transducer mechanisms, such as electrochemical, field-effect transistors (FET), and piezoelectric. These sensors are widely used in various applications, including medical diagnostics, environmental monitoring, and food safety testing, due to their high sensitivity, specificity, and the potential for miniaturization and integration into portable devices.

Electrochemical aptasensors use electrodes as transducers to measure biomolecular interactions by detecting changes in surface charge [12]. In other words, the binding of a target molecule to the aptamer on the electrode surface will induce redox reactions that result in charges on electrodes that do modify the current, voltage, or impedance of an electrical circuit, which are then used to identify and quantify the analyte. According to that, electrochemical aptasensors can be broadly classified into amperometric and voltametric, or potentiometric and impedimetric aptasensors. The first ones, amperometric and voltametric aptasensors, operate based on the measurement of the current at an electrode surface as a result of the electrochemical reaction produced by the interaction between the analyte and aptamer [25]. The difference between amperometric and voltametric aptasensors is in how voltage is applied and current changes are measured. While voltametric sensors measure the variations in the current signal over a variable voltage range as an indication of analyte concentration, amperometric aptasensors measure the current changes over time at a fixed voltage [12]. On the other hand, potentiometric aptasensors work by measuring the voltage difference between two electrodes, being commonly used due to their simple operation, portability, and low cost [25]. Finally, impedimetric aptasensors measure changes in electrical impedance resulting from the binding of the target molecules to the aptamer-functionalized surfaces [12]. In general, electrochemical aptasensors have low detection limits, high sensitivity, real-time detection, robustness, and potential for miniaturization [12,25,105]. However, these sensors may include optimizing electrode surface functionalization, ensuring specificity, and addressing potential interference from complex sample matrices, which can be a challenge.

Another transduction mechanism commonly used is the one that employs a field-effect transistor (FET). Any adsorption of biomolecules on the transistor’s channel surface causes changes in the electric field that modulates the gate potential, resulting in a change in the drain current [104,109]. This type of aptasensor offers many advantages such as miniaturization, low cost, large-scale integration capability with the existing manufacturing process, as well as being label-free, offering rapid detection, and highly sensitive detection of analytes. Like the electrochemical aptasensors, some issues related to optimization of surface functionalization may appear. Furthermore, the sensitivity of this kind of transducer mechanism can be affected by pH and ionic strength [109].

Piezoelectric transducers can also be used as detection mechanisms for electrical aptasensors, in which the piezoelectric effect, i.e., the generation of an electric charge when mechanical stress or pressure is applied to specific materials, is used [25]. Binding of target molecules to aptamers immobilized on the surface causes mechanical stress changes, which result in frequency shifts that can be measured electrically, typically using a quartz crystal microbalance device. Piezoelectric aptasensors have some disadvantages and challenges associated with their use, such as the cost and sensitivity to environmental factors. Despite these disadvantages, piezoelectric aptasensors remain a valuable tool as a transducer mechanism by offering high sensitivity, label-free detection, real-time monitoring, and versatility [104].

Overall, electrical aptasensors play a crucial role in advancing biosensing technologies due to their high sensitivity, specificity, and real-time monitoring capabilities. Their versatility makes them valuable tools for researchers and professionals in various fields seeking accurate and rapid detection methods.

#### 3.2.3. Magnetic Detection

Magnetic aptasensors are a class of biosensors that utilize magnetic fields for the detection and quantification of specific target molecules. In particular, magnetoresistive sensors are devices that measure the alterations in electrical resistance of a material in response to an applied magnetic field [110], and their development is significantly increasing towards bio-applications as well as for point-of-care diagnostics [7]. This type of aptasensor had some advantages, such as low power consumption, high sensitivity, portability, and low cost [110]. However, some sensing approaches may need more complex signal processing in order to convert the resistance changes into meaningful data [110].

## 4. Applications of Aptasensors: Point-of-Care Diagnosis

Since the first development of the aptamer sequence, a variety of aptasensors have been reported with enormous attention on their possible applications in clinical diagnosis [111]. Most of the aptasensors are focused on the detection of biomarkers related to infectious diseases, cancer, CVDs, metabolomic diseases, and neurological disorders such as dementia, and in particular Alzheimer’s disease (AD). Table 4 summarizes some of the aptasensors already developed for the detection of some important biomarkers in clinical diagnosis applications.

### 4.1. Aptasensors for Infection Biomarkers

Aptasensors have been developed for the specific detection of pathogenic bacteria, notably *E. coli* O157:H7. In a study by Chung et al. [112], a real-time and continuous aptasensor for *E. coli* detection was established. This aptasensor employed target-specific aptamer-conjugated fluorescent nanoparticles and an optofluidic particle-sensor platform.

In the context of viral infections, aptamers have been investigated for their capacity to target viruses such as AIV, HIV, and coronaviruses, including SARS-CoV-2, responsible for the COVID-19 pandemic. Lum et al. [113] and Chan et al. [114] both developed aptasensors for the specific detection of H5N1 AIV. The key distinction between these two aptasensors lies in their detection principles. Lum et al. used impedimetric technology, achieving a limit of detection (LOD) of 0.0128 haemagglutinin units (HAU) [113], while Chan et al.’s aptasensor was based on FET technology, achieving a LOD of 5 pM [114].

For the detection of p24-HIV protein, Gogola et al. [115] designed a label-free aptasensor that employed graphene quantum dots (QDs) as an electrochemical signal amplifier. Under optimal experimental conditions, this aptasensor exhibited a linear correlation between the analytical signal and the logarithm of p24-HIV concentration, spanning from 0.93 ng/mL to 93 mg/mL, with a LOD of 51.7 pg/mL.

In the realm of SARS-CoV-2 detection, several aptasensors have been developed. An electrochemical aptasensor was created for highly sensitive SARS-CoV-2 antigen detection, using an aptamer-binding-induced multiple-hairpin assembly strategy for signal amplification [116]. This aptasensor designed for SARS-CoV-2 spike protein detection displayed a broad linear range from 50 fg/mL to 50 ng/mL and a low LOD of 9.79 fg/mL [116]. Cennamo et al. [117] developed an optical fibre SPR-based aptasensor with a LOD of approximately 37 nM. Chen et al. [118] developed an aptasensor capable of detecting SARS-CoV-2, based on SERS, achieving a LOD of less than 10 PFU (plaque-forming unit)/mL within 15 min [118].

### 4.2. Aptasensors for Cancer Biomarkers

Cancer is a global health challenge, with early detection playing a crucial role in improving patient outcomes [2]. Aptasensors offer a promising path for the sensitive and selective detection of cancer biomarkers [119].

PSA serves as a fundamental biomarker in the diagnosis of prostate-related conditions, particularly prostate cancer. PSA is usually measured in nanograms per millilitre of blood, and values above PSA 2.5 or 4 ng/mL are considered abnormal for men aged 40–50 or 60 years old or more, respectively [120]. As such, several aptamer-based biosensors have been created, including those detailed by Jolly et al. [121] and Liu et al. [122]. In the former, the DNA aptamer sequence was used as a capture layer. The process involved incubating Au-electrodes with thiol-terminated sulfobetaine and mercaptoundecanoic acid, enabling the detection of PSA at concentrations below 1 ng/mL [121]. Meanwhile, in the latter PSA aptasensor [122], aptamer immobilization was achieved through avidin-biotin surface chemistry. This aptasensor demonstrated a linear response to PSA within the range of 0.25 to 200 ng/mL, and a LOD of 0.25 ng/mL [122].

Elevated levels of HER2 are associated with early recurrence and metastatic disease in human breast cancer [123], whereas a median value in serum of 12.2 ng/mL has been considered to be normal [124]. Shen et al. [125] devised a biosensor based on aptamer technology for detecting HER2. In this system, an aptamer served a dual role as a recognition ligand and a signal-generating reporter, using an aptamer–analyte–aptamer sandwich format. The biosensor exhibited a linear response to HER2 concentrations ranging from 1 pg/mL to 100 pg/mL, with a LOD of 0.047 pg/mL [125]. Moreover, Bezzera et al. [54] presented an electrochemical aptasensor designed for HER2 in which the aptamer was immobilized through electrostatic adsorption onto a custom-made screen-printed electrode modified with poly-L-lysine. This aptamer-based biosensor displayed a linear response in the concentration range of 10 to 60 ng/mL and a LOD of 3.0 ng/mL [54]. Apart from electrical detection techniques, an optical sensing approach was also employed for HER2 detection. For instance, Ranganathan et al. [126] developed assays exploring two different methods using HER2-binding aptamers and AuNPs. The first method involved a solution-based adsorption–desorption colorimetric approach, in which aptamers were adsorbed onto the surface of AuNPs. In the presence of HER2, the colour of the solution changed from red to blue. In contrast, the second method used an adsorption–desorption colorimetric lateral flow assay approach in which biotin-modified aptamers were adsorbed onto the AuNPs’ surface in the absence of HER2. Both approaches were tested, resulting in LODs of 10 nM and 20 nM, respectively [126]. Within the realm of optical sensing methods, Loyez et al. [127] employed SPR aptasensors as a detection tool for HER2, achieving a LOD of 20 g/mL.

In addition to HER2, the overexpression of osteopontin (OPN), a protein found in various bodily fluids, is also recognized as a biomarker related to breast cancer evolution and metastasis [128]. The reported normal median plasma OPN levels are highly variable, ranging from 31 ng/mL to 200 ng/mL [129]. Meirinho et al. [130] employed the SELEX process to identify a DNA aptamer for human OPN, which they used as a biomolecular recognition tool in a biosensor. The biosensor exhibited a linear response within the concentration range of 25 to 100 nM, with a LOD of 1 nM. Furthermore, the same research group had previously developed another assay for OPN [131]. The primary distinction between these two aptasensors lay in the type of aptamer employed. The first one employed a DNA aptamer, while the second one used an RNA-based aptamer. The RNA aptasensor could detect human OPN with a LOD of 3.7 nM [131].

MUC1, a glycoprotein present on the surfaces of epithelial cells, serves as a valuable biomarker for the early detection of cancers [132]. Various aptamer-based biosensors have been developed for the detection of MUC1. For instance, Wen et al. [133] introduced an electrochemical aptasensor using voltammetry. This aptasensor exhibited a good linear correlation within the range of 10 pM to 1 μM, and a LOD of 4 pM [133]. Additional studies [132,134,135] have reported the development of an aptasensor employing two optical detection methods, fluorescence and chemiluminescence. Cheng et al. [132] and Chen et al. [134] developed aptamer-based biosensors capable of targeting and quantifying MUC1 through fluorescence. Cheng et al. [132] designed a three-component DNA system incorporating fluorescent semiconductor QDs to selectively detect the MUC1. In the absence of MUC1, strong fluorescence is observed. However, in the presence of MUC1, a significant reduction in fluorescence intensity occurred due to the binding between MUC1 and the aptamer. The biosensor displayed a linear response to MUC1 concentrations within the range of 0 to 2.0 µM, and a LOD of 250 nM. Similarly, Chen et al. [134] used QDs as nanoscale signal reporters for homogeneous visual and fluorescent detection of MUC1, achieving a LOD of 0.15 fg/mL. Concerning chemiluminescence aptasensors, Ma et al. [135] introduced a microfluidic paper-based analytical device designed for the sensitive detection of peptides. In this study, MUC1 served as the model peptide, and the assay exhibited a LOD of 8.33 pM, operating effectively within a concentration range spanning from 25 pM to 50 nM.

### 4.3. Aptasensors for Cardiovascular Biomarkers

Aptasensors have emerged as promising tools in the detection of CVDs by targeting specific biomarkers associated with these conditions. One prominent biomarker frequently investigated in CVD diagnostics is cardiac troponin, specifically cTnI. To detect cTnI, Lopa et al. [136] developed an aptasensor using a titanium metal substrate adorned with gold AuNPs and modified with a self-assembled monolayer of a thiol-functionalized DNA aptamer. This aptasensor exhibited a linear response to cTnI concentrations within the range of 1 to 1100 pM, achieving a LOD of 0.18 pM, making it a highly sensitive tool for early CVD diagnosis [136]. Moreover, Jo et al. [55] designed an electrochemical aptasensor for cTnI detection based on square wave voltammetry, using ferrocene-modified silica nanoparticles. This aptasensor showed excellent analytical performance, featuring a linear range from 1 to 10,000 pM in buffer solutions and a LOD of 1.0 pM [55].

Myoglobin is a protein found in muscle tissues and is released into the bloodstream in case of muscle lesion, such as a myocardial infarction [137]. To monitor myoglobin levels in blood serum, various diagnostic assays, and sensors, such as aptasensors, have been developed. An example of that is the study published by Taghdisi et al. [138], in which a novel electrochemical aptasensor was designed using a Y-shaped dual-aptamer structure. This aptasensor achieved a LOD as low as 27 pM.

Sinha et al. reported the development of a sensitive aptamer-based planar hall magnetoresistive biosensor for the detection of human α-thrombin. An aptamer–thrombin–aptamer sandwich assay was then used, showing a linear response within the range of 86 pM to 8.6 µM, and a LOD of 86 pM [7]. On the other hand, Deng et al. [139] developed a combined piezoelectric and SERS-based aptasensor for the detection of thrombin. The LOD for thrombin was 0.1 μM, and in the concentration range of 0.1 to 1.0 μM, a good linear correlation was obtained [139].

Furthermore, CRP and IL-6, have been recognized as important inflammatory biomarkers associated with CVD risk. In a study conducted by Pultar et al. [140], the focus was on detecting CRP using an aptamer-based biosensor. This aptasensor employed an optical detection method, using a labelled secondary antibody in a sandwich format to detect bound CRP. Notably, the aptasensor displayed a linear response over a wide concentration range, spanning from 10 µg/L to 100 mg/L, and a LOD of 10 µg/L [140]. Another development was a disposable electrochemical aptasensor designed by Centi et al. [141] for CRP screening. This aptasensor employed a sandwich format, in which an RNA-based aptamer was linked to a monoclonal antibody and alkaline phosphatase. The LOD, as determined in CRP-free serum, was reported to be 0.2 mg/L. Furthermore, Wang et al. [142] introduced an electrochemical aptasensor for CRP detection, using functionalized silica microspheres. This aptasensor showed a wide linear range of 0.005 ng/mL to 125 ng/mL for CRP detection, and a low LOD at 0.0017 ng/mL. Regarding, IL-6, Tertis et al. [71] have recently described a label-free electrochemical aptasensor for the sensitive recognition of IL-6 in human serum. The IL-6 specific aptamer was immobilized using sulphur–gold bonding and monitored using voltammetry and electrochemical impedance spectroscopic-based techniques. The results showed that IL-6 could be detected in an extensive linear range from 1 pg/mL to 15 µg/mL with a LOD of 0.33 pg/mL.

### 4.4. Aptasensors for Metabolic Biomarkers

Aptasensors have been developed to detect metabolomic biomarkers, such as HbA1c [83,143], GHSA, RBP4 [86,144], and others associated with metabolic diseases.

Chang et al. [143] developed an integrated microfluidic system for the measurement of both HbA1c and Hb, by using an aptamer–antibody assay on magnetic beads. This system showed a linear range from 0.65 to 1.86 g/dL for HbA1c and 8.8 to 14.9 g/dL for Hb. Similarly, Li et al. [83] reported an aptamer-based microfluidic system which simultaneously performed two parallel assays for the detection of Hb and HbA1c.

In a study published by Belsare et al. [145], two aptamer-based lateral flow assays were developed to measure GHSA and serum albumin in their relevant concentration ranges [145]. Herein, both assays used AuNP-based colorimetry, and the dynamic ranges for both assays were 3 to 20 mg/mL and 20 to 50 mg/mL, respectively, which are both physiologically relevant. The LOD was calculated to be 0.8 mg/mL and 1.5 mg/mL, respectively [145].

Moabelo et al. [144] showed an aptasensor using AuNPs to detect RBP4. The aptamers designed for RBP4 recognition bonded to the protein, causing them to disengage from the AuNPs’ surface. This detachment resulted in the aggregation of AuNPs, a phenomenon that induces a visible shift in the colour of the AuNP solution from red to purple or blue. The assay had a LOD of 90.76 ± 2.81 nM. In a previous study, Lee et al. [86] developed an aptamer-based SPR biosensor that could be used to sense RBP4 in serum samples with a linear range and a LOD for RBP4 of 0.2 to 0.5 µg/mL and 75 nM, respectively.

### 4.5. Aptasensors for Neurological Biomarkers

Neurological disorders, encompassing conditions like AD and Parkinson’s disease, often require precise and early diagnosis for effective management [37].

Aptamers have been developed to target specific biomarkers like Aβ peptides and Tau protein for AD diagnosis. Lu et al. [146] introduced an innovative aptasensor capable of simultaneously detecting these two distinct AD disease biomarkers. They used the exceptional photoluminescent properties of QDs in the development of a fluorescence resonance energy transfer aptasensor. This groundbreaking approach resulted in a linear response across concentration ranges of 100 to 2000 pM for Aβ and 50 to 1500 pM for Tau protein, featuring low LODs of 50 nM and 20 nM, respectively [146]. Furthermore, Zhang et al. [147] proposed a simple yet sensitive aptasensor that focused on the selective detection of Aβ oligomers through amperometric response alterations. This aptasensor showed an extensive concentration detection range spanning from 0.1 pM to 1500 nM, and a remarkable sensitivity with a LOD at the femtomolar level [147]. In a study conducted by Tao et al. [148], an electrochemical aptasensor was designed for detecting Tau-381. Under optimal conditions, the increment of differential pulse voltammetry signal increased linearly with the logarithm of Tau-381 concentration in the range from 1.0 pM to 100 pM, and the LOD was 0.70 pM [148]. Similarly, Shui et al. [149] reported a novel aptamer–antibody sandwich assay with electrochemical transduction for the detection of Tau-381 in human serum. The sensor exhibited a wide range of detection between 0.5 pM and 100 pM, with a LOD of 0.42 pM for Tau-381.

PD, characterized by the loss of dopamine-producing neurons, benefits from aptasensors that can detect dopamine or α-syn, a protein associated with the disease [37]. Kim et al. [150] developed a point-of-care platform for the diagnosis of PD using a conductive polymer, an aptamer receptor on a screen-printed electrode, and electrochemical impedance spectroscopic analysis, showing a remarkable low detection limit of 10^−9^ nM levels.

An important biomarker is dopamine, a neurotransmitter that plays a significant function in human mental and physical processes. Disturbances in dopamine levels may lead to neurodegenerative diseases, such as PD, and Huntington’s disease (HD) [111]. In a study by Dalirirad et al. [151], a lateral flow assay using DNA aptamer-based sensing was devised for dopamine detection in urine. The methodology hinges on the dissociation of a duplex aptamer in the presence of dopamine, causing conformational changes and subsequent release from the capture component. When complementary DNA in the test line hybridizes with the conjugated AuNP-capture DNA, a visible red band forms, with its intensity directly proportional to dopamine concentration. The achieved LOD was 10 ng/mL [151]. In a study conducted by Hu et al. [152], a DNA aptamer’s capability to capture dopamine molecules, prompting distinct conformational changes, was showcased at the surface of optical fibres. Their aptasensor relied on amplifying the modulation of the surface refractive index along the fibre surface, allowing for real-time measurement of dopamine concentrations by monitoring surface plasmon resonance signals. This sensor exhibited a linear response within a dopamine concentration range spanning from 0.1 pM to 10^4^ pM, and a LOD of 0.1 pM [152].

### 4.6. Aptasensors for Continuous Biomarker Detection

Continuous biomarker monitoring is a crucial requirement for effective point-of-care applications. Aptamers offer a unique advantage in this regard due to the reversibility of conformation, allowing for uninterrupted measurements. For instance, Ferguson et al. [153] conducted a study that explored the aptamer’s conformational dynamics, leading to the development of a real-time aptamer-based biosensor designed for continuous monitoring of circulating drugs. In this approach, the biosensor chip is directly interfaced with the patient’s bloodstream, and the aptamer probe is anchored to a gold electrode. The binding event of the target analyte induces a reversible conformational change in the aptamer probe, resulting in an increased electron transfer rate, leading to a measurable change in electric current. This aptasensor demonstrated a LOD of 10 nM in buffer solutions, with a dynamic range of 0.01 to 10 mM [153].

Additionally, Swensen et al. [154] reported the development of an electrochemical aptamer-based biosensor to achieve continuous and real-time monitoring of cocaine in undiluted and unmodified blood serum [154]. Finally, Liu et al. [155] also reported an electrochemical aptasensor; however, instead of continuously detecting cocaine, this aptasensor allowed for the continuous detection of MUC1, demonstrating a linear response within the range of 10 nM to 100 nM, and a LOD of 13 nM [155].
biomedicines-11-03201-t004_Table 4Table 4Aptasensors for biomarker detection.TargetDetection MethodLinear RangeLODSample TestedReferenceInfectious Diseases*E. coli*Fluorescence---[112]H5N1 AIVImpedimetric-0.0128 HAUBuffer[113]FET10 pM to 100 nM5 pMBuffer[114]p24 HIVElectrochemical0.93 ng/mL to 96 mg/mL51.7 pg/mLBuffer[115]Electrochemical50 fg/mL to 50 ng/mL9.79 fg/mL-[116]SARS-CoV-2SPR-37 nMBuffer[117]SERS-10 PFU/mL-[118]CancerPSAImpedimetric-1 ng/mLBuffer[121]Voltametric0.25 to 200 ng/mL0.25 ng/mLSerum Sample[122]HER2Impedimetric1 to 100 pg/mL0.047 pg/mLSerum Sample[125]Voltametric10 to 60 ng/mL3.0 ng/mLSerum Sample and Buffer[54]Colorimetric0 nM to 99 nM0 nM to 50 nM10 nM20 nMBuffer[126]SPR-20 g/mLBuffer[127]OPNVoltametric25 to 100 nM1 nMSynthetic Human Plasma[130]Voltametric25 to 200 nM3.7 nM-[131]MUC1Voltametric10 pM to 1 µM4 pMBuffer[133]Fluorescence0 to 2.0 µM 250 nM-[132]Fluorescence-0.15 fg/mL-[134]Chemiluminescence25 pM to 50 nM8.33 pMSerum Sample[135]CVDscTnIVoltametric1 to 1100 pM0.18 pM-[136]Voltametric1 to 10,000 pM1.0 pMSerum Sample[55]MyoglobinVoltametric100 pM to 40 nM27 pMBuffer[138]ThrombinMagnetic86 pM to 8.6 µM86 pMBuffer[7]Piezoelectric and SERS0.1 to 1.0 µM0.1 µMBuffer[139]CRPFluorescence 10 µg/L to 100 mg/L10 µg/LBuffer[140]Voltametric-0.2 mg/LSerum Sample[141]Voltametric0.005 to 125 ng/mL0.0017 ng/mLBuffer[142]IL-6Voltametric and Impedimetric 1 pg/mL to 15 µg/mL0.33 pg/mLBuffer[71]Metabolomic DiseasesHbA1c and HbChemiluminescence0.65 to 1.86 g/dL8.8 to 149 g/dL-Blood[143]Fluorescence--Blood[83]Glycated AlbuminColorimetric3 to 20 mg/mL0.8 mg/mLSerum Sample[145]Serum AlbuminColorimetric20 to 50 mg/mL1.5 mg/mLSerum Sample[145]RBP4Colorimetric-90.76 ± 2.81 nMBuffer[144]
SPR0.2 to 0.5 µg/mL75 nMSynthetic Human Plasma[86]Neurological DiseasesAβFluorescence100 to 2000 pM50 nMBuffer[146]Amperometric0.1 pM to 1500 nM6.5 fMBuffer[147]Tau proteinFluorescence50 to 1500 pM20 nMBuffer[146]Tau-381Voltametric1.0 to 100 pM0.70 pMSerum Sample[148]Voltametric0.5 to 100 pM0.42 pMBuffer[149]α-synImpedimetric10^−8^ to 0.1 nM10^−9^ nM-[150]DopamineColorimetric-10 ng/mLBuffer[151]SPR0.1 pM to 10^4^ pM 10^4^ pMSerum Sample[152]Continuous MonitoringCirculating DrugsVoltametric0.01 to 10 mM10 nMBuffer [153]CocaineElectrochemical--Serum Sample[154]MUC1Voltametric and Impedimetric10 to 100 nM13 nMBuffer[155]AIV—avian influenza virus; HAU—haemagglutinin units; FET—field-effect transistor; HIV—human immunodeficiency viruses; SPR—surface plasmon resonance; SERS—surface-enhanced Raman scattering; PFU—plaque-forming unit; PSA—prostate-specific antigen; HER2—human epidermal growth factor response; OPN—osteopontin; MUC1—Mucin-1; cTnI—cardiac troponin I; CRP—C-reactive protein; IL-6—interleukine-6; HbA1c—glycated haemoglobin A1c; Hb—haemoglobin; RBP4—retinol binding protein 4; Aβ—amyloid-beta; α-syn—α-synuclein.

## 5. Conclusions and Future Directions

In the realm of sensor applications, aptamers have emerged as highly promising alternatives to antibodies, featuring unique properties that set them apart. Their exceptional specificity and affinity for target molecules, combined with the cost-effective and rapid selection process, make them immensely appealing for healthcare and other fields.

The ongoing exploration of aptamers and their corresponding aptasensors in the field of clinical diagnostics shows exciting possibilities for enhancing disease detection, monitoring, and patient care. However, further research is imperative to expand the repertoire of aptamers targeting an extensive array of biomarkers associated with various diseases.

Aptamers are often developed under controlled laboratory conditions, and the majority of the aptasensors presented were tested in buffer systems instead of complex clinical samples, such as human serum, whole blood, urine, or other tissues. It may indicate that the performance of aptamers in complex biological environments may not be fully understood. So, future research should focus on evaluating the stability and functionality of aptamers in complex biological matrices, advancing their applicability in real-world scenarios. Long-term stability of aptasensors and optimal storage conditions are crucial for their application but are not yet fully addressed, so further research is needed.

Furthermore, current aptasensors often target a single analyte, limiting their applicability for complex analyses. In this way, future developments should aim to create aptasensors with the capability to simultaneously detect multiple analytes, meeting the demand for versatile and comprehensive biosensing platforms.

Overall, this comprehensive literature review underscores the key role of aptamers and aptasensors in biosensing, emphasizing their versatile nature as molecular recognition tools and clinical potential for point-of-care diagnosis over a wide range of domains, including detection of infection, cancer, CVDs, and metabolic and neurological diseases. The focus on point-of-care applications distinguishes this work, providing crucial insights into the translational potential of aptamers for real-world diagnostics. In other words, this review paper hopes to contribute to the translational potential of aptasensors from research to practical diagnostics.

## Figures and Tables

**Figure 1 biomedicines-11-03201-f001:**
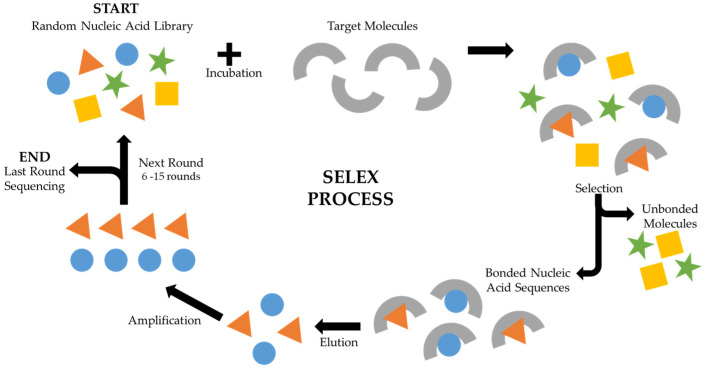
Schematic representation of the SELEX process. The SELEX process starts with the incubation of the random nucleic acid library with the target. The unbound nucleic acids are removed before the bonded nucleic acid sequences are eluted and amplified. These sequences serve as the initial library of the next round of SELEX. Typically, 5 to 16 SELEX rounds of evermore stringent selections are performed before sequencing of the most target-affined nucleic acid sequence, which then follows to production.

**Figure 2 biomedicines-11-03201-f002:**
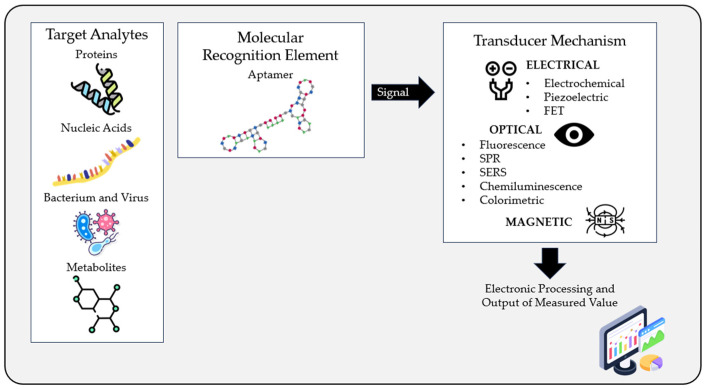
Schematic diagram showing the operating principle of an aptasensor. The target molecule binds to the aptamer, generating a signal that is detected by a transducer element. This signal is then processed, and a measurable value is obtained. FET—field-effect transistor; SPR—surface plasmon resonance; SERS—surface-enhanced Raman scattering.

**Figure 3 biomedicines-11-03201-f003:**
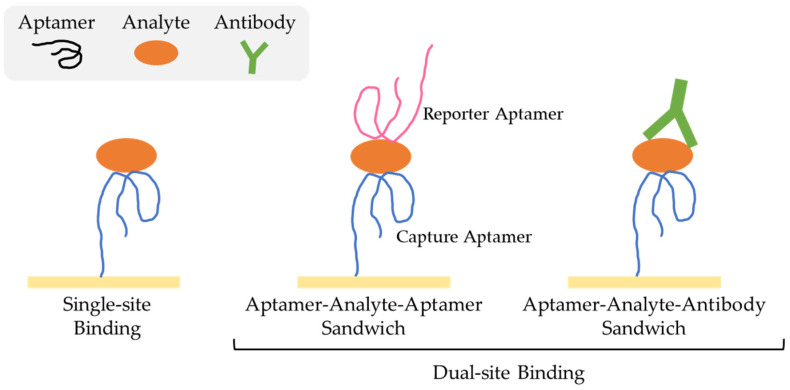
Different types of assay configurations.

**Figure 4 biomedicines-11-03201-f004:**
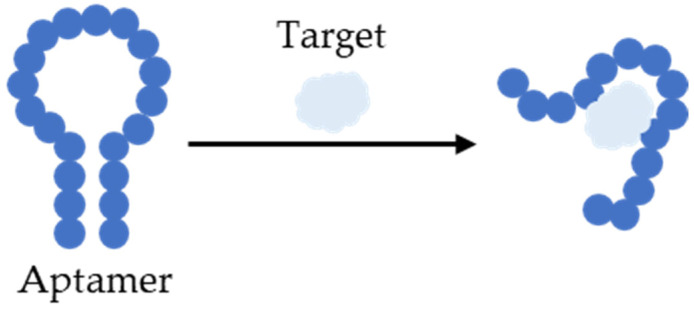
Aptamer switch conformation when binding to its target.

**Table 1 biomedicines-11-03201-t001:** Comparation between aptamers and antibodies, adapted from the literature [27,28,29,31,32].

Features	Aptamers	Antibodies
Molecular Weight	5 to 15 kDa	150 to 170 kDa
Selection Process	SELEX	Produced via the immune response B-cell maturation
Generation Time	Months	Several months
Production Variability	Lower	Higher
Production Scalability	Scalable	Limited scalability
Shelf Life	Long	Short
Modifications	Easily modified for improved properties	Limited modification options
Stability	Stable under various conditions	Sensitive to pH, temperature, pressure, and denaturation
Cost	Lower	Higher
Ethical Concerns	None: production and development are chemical reactions	Ethical challenges: production and development are highly dependent on living animals
Nuclease Susceptibility	Present	Absent

**Table 2 biomedicines-11-03201-t002:** Summary of aptamers identified and their dissociation constant and type.

Target		Medical Condition	Type	Dissociation Constant (K_d_)	Reference
Pathogenic Microorganisms	*Escherichia coli* O157:H7	Gastroenteritis	DNA	10.30 nM	[39]
			DNA	107.6 ± 67.8 pM	[38]
	*Salmonella typhimurium*	Gastroenteritis	DNA	6.33 ± 0.58 nM	[40]
			DNA	nM to µM range	[41]
	*Salmonella enteritidis*	Gastroenteritis	DNA	nM to µM range	[41]
	*Mycobacterium tuberculosis*	Tuberculosis	DNA	nM range	[42]
	Avian Influenza Virus	Respiratory tract infection	DNA	nM range	[97]
			RNA	-	[98]
			DNA	55.14 ± 22.40 nM	[99]
	HIV	Acquired Immunodeficiency Syndrome	RNA	-	[43]
			RNA	0.15 nM	[44]
			DNA	-	[45]
	SARS-CoV-2	Respiratory tract infection	RNA	pM range	[46]
Cancer Biomarkers	Nucleolin	Gastric and non-small-cell lung cancer	DNA	73 nM	[47]
	Tenascin	Colorectal, breast, and ovarian cancer	DNA	-	[48]
	PSA	Prostate cancer	RNA	-	[49]
	Mucin 1	Breast cancer	DNA	-	[50]
	Annexin A2	Tumour progression	RNA	nM range	[51]
	MMP-9	Tumour progression	RNA	nM to µM range	[52]
			RNA	20 nM	[53]
	HER2	Breast cancer	RNA	-	[54]
Cardiac Biomarkers	Cardiac Troponin I	Heart attack	DNA	pM range	[55]
			DNA	nM range	[56]
			DNA	19.8 nM	[57]
			DNA	nM range	[58]
			DNA	nM range	[59]
			DNA	-	[60]
	Cardiac Troponin T	Heart attack	DNA	-	[61]
			DNA	nM range	[62]
			DNA	43 nM	[63,64]
	Myoglobin	Skeletal and heart muscle damage	DNA	nM range	[65]
			DNA	-	[66]
			DNA	65 pM	[67,68]
	Creatinine Kinase	Skeletal and heart muscle damage	DNA	nM range	[69]
Inflammatory Biomarkers	Interleukin-6	Inflammation	DNA	µM range	[70]
			DNA	-	[71]
			DNA	nM range	[72]
			DNA	-	[73]
	C-Reactive Protein	Inflammation	DNA	nM range	[74]
			DNA	16.2 nM	[75]
			DNA	3.59 nM	[76]
			RNA	125 nM	[77]
			RNA	2.25 nM	[78]
			DNA	nM range	[79]
	Tumour Necrosis Factor Alpha	Inflammatory and autoimmune diseases	DNA	8 nM	[80]
			DNA	nM range	[81]
			DNA	0.35 nM	[82]
Metabolic Biomarkers	HbA1c	Diabetes mellitus	DNA	7.6 ± 3.0 nM	[83,84]
	GHSA	Diabetes mellitus	DNA	nM range	[85]
	RBP4	Obesity	DNA	0.20 ± 0.03 µM	[86]
	Mature Adipocyte Cell Line	Obesity	DNA	17.8 ± 5.1 nM	[87]
	Vaspin	Obesity	DNA	µM range	[88]
Neurological Biomarkers	Amyloid-β	Alzheimer’s disease	DNA	nM range	[89]
	Amyloid-β 1-40	Alzheimer’s disease	RNA	µM range	[90]
	Tau-381	Alzheimer’s disease	DNA	190 nM	[91]
	Tau-441	Alzheimer’s disease	DNA	28 nM	[92]
	Dopamine	Dopamine	RNA	2.8 µM	[93]
	α-synuclein	Parkinson’s disease	DNA	-	[94]
	Huntingtin	Huntington’s disease	DNA	-	[95]

HIV—human immunodeficiency viruses; PSA—prostate-specific antigen; MMP-9—matrix metalloprotease-9; HER2—human epidermal growth factor response; HbA1c—glycated haemoglobin A1c; GHSA—glycated human serum albumin.

**Table 3 biomedicines-11-03201-t003:** Summary of the transducer mechanisms used in aptasensors and their advantages and disadvantages.

Transducer		Sensing Mechanism	Advantages	Disadvantages
Optical	Fluorescence	Measures alterations in emitted fluorescent signals	High sensitivity Low detection limits Real-time monitoring Cost-effectiveness Diverse fluorophore options	Background fluorescence Photobleaching of fluorophore Interference from complex samples
	Chemiluminescence	Measures light emission from chemical reactions	High sensitivity Real-time monitoring Low background signal Simplicity	Requires chemical substrates Limited signal duration
	SPR	Measures alterations in the refractive index	High sensitivity Real-time monitoring Label-free detection Simplicity	Cost Single-target detection
	SERS	Measures Raman signals	High sensitivity Label-free detection Multiplexing	Limited reproducibility
	Colorimetric	Measures colour alterations	Simplicity Cost-effectiveness Easy interpretation	Limited sensitivity
Electrical	Electrochemical	Measures alterations in electrical signals	High sensitivity Low detection limits Real-time monitoring Label-free detection Robustness Miniaturization	Optimized surface functionalization required Potential interference from complex samples
	FET	Measures alterations in electrical conductivity upon target binding	High sensitivity Cost-effectiveness Label-free detection Rapid detection Large-scale integration Miniaturization	Optimized surface functionalization required Sensitive to pH and ionic strength
	Piezoelectric	Measures alteration in mechanical stress	High sensitivity Real-time monitoring Label-free detection Versatility	Sensitive to environmental factors
Magnetic	Magnetoresistive	Measures the alterations in electrical resistance of a material in response to an applied magnetic field	Low-power consumption High sensitivity Portability Low-cost	May need more complex signal processing

SPR—surface plasmon resonance; SERS—surface-enhanced Raman scattering; FET—field-effect transistor.

## Data Availability

Not applicable.

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
