# Peer review of "Nucleic Acid Aptamer-Based Biosensors: A Review"

_biomedicines, 2023, doi:10.3390/biomedicines11123201_

Round 1
Reviewer 1 Report
Comments and Suggestions for Authors
Dear authors,
I appriciate your efforts to cover the enourmous range of the papers devoted to aptasensors for biomarkers. Your review is structured and written with good and comprehensive language.
But behind this extensive infromation I do not see a goal of your review. In the current version it is a recitation of other researches. It lacks some new ideas and outlines, which distict a good review from poor one.
So my main comment is the following. Please, think on the goals of your work, and find some systematization, conclusions and outlines derived from the literature. This may be trends, common problems, some experimental issues which were not obvious before. This will enrich your paper with scientific contribution and novelty.
Besides, I have some minor comments listed below:
1. In line 83 of maintext you say SELEX is tipically 6 to 15 cycles, in caption to figure 1 - 6 to 16.
2. I recommend fo indicate that identification of candidate aptamers and thorough investigation of binding charachteristics of an aptamer is very important for aptamer discovery.
3. You indicate the benefits of aptamers but do not discuss the drawbacks.
4. In 171-172 there is a misleading sentence on stucture switch of aptamers. Structure switch upon target binding is not a default feature of aptamers, it is a possible variant.
5. Figure 2 displayes that aptamers can detect nucleic acids. Please provide some references to support this idea. Are there really some aptamers which are affinity ligands to nucleic acids besides complementarity?
6. I see an issue in the section on optical sensors. Chemiluminescence, HRP and DNA-zymes are distinct from other optical detection due to signal amplification by enzymatic activity.
Reviewer 2 Report
Comments and Suggestions for Authors
The review entitled "Nucleic Acid Aptamer-Based Biosensors: A Review" presents a comprehensive overview of aptamers used for biosensing.
The review is of interest to the community working on aptamer-based systems and is extensive in its coverage of the clinical applications of this type of biosensor.
I do think, however, that a few minor additions would provide a welcome complement to this study:
- The review focuses on clinical applications, but I think it would be pertinent to mention surreptitiously that the applications go beyond the biomedical realm.
- One of the most important aspects and current challenges of biosensors is their ability to work directly in biological media (such as blood, for example). It would be helpful to mention this aspect and its current status, perhaps in the outlook or, in Table 4, to add a column with the medium in which the studies were carried out (buffer or blood).
- One of the challenges for point-of-care applications is to be able to measure biomarkers continuously (especially those associated with cardiovascular risk, whose concentration evolves over short time scales and for which end-point measurements are not sufficient). The use of aptamers enables such continuous measurements, thanks to the reversibility of some of their properties (e.g. conformation, binding). I think it would be interesting to mention this, illustrating it with one or two examples (such as :10.1126/scitranslmed.3007095)
Considering some of the above suggestions, I agree with the publication of this review in Biomedicines.
Round 2
Reviewer 1 Report
Comments and Suggestions for Authors
With this changes the manuscript can be accepted.